# Can natural language processing models extract and classify instances of interpersonal violence in mental healthcare electronic records: an applied evaluative study

Riley Botelle [1], Vishal Bhavsar,[2] Giouliana Kadra-Scalzo [3], Aurelie Mascio,[3] Marcus V Williams,[1] Angus Roberts,[4,5] Sumithra Velupillai,[3] Robert Stewart[3,6]

For numbered affiliations see end of article.

**Correspondence to**
Dr Riley Botelle;
riley.botelle@gmail.com

## ABSTRACT

**Objective** This paper evaluates the application of a natural language processing (NLP) model for extracting clinical text referring to interpersonal violence using electronic health records (EHRs) from a large mental healthcare provider.

**Design** A multidisciplinary team iteratively developed guidelines for annotating clinical text referring to violence. Keywords were used to generate a dataset which was annotated (ie, classified as affirmed, negated or irrelevant) for: presence of violence, patient status (ie, as perpetrator, witness and/or victim of violence) and violence type (domestic, physical and/or sexual). An NLP approach using a pretrained transformer model, BioBERT (Bidirectional Encoder Representations from Transformers for Biomedical Text Mining) was fine-tuned on the annotated dataset and evaluated using 10-fold cross-validation.

**Setting** We used the Clinical Records Interactive Search (CRIS) database, comprising over 500 000 de-identified EHRs of patients within the South London and Maudsley NHS Foundation Trust, a specialist mental healthcare provider serving an urban catchment area.

**Participants** Searches of CRIS were carried out based on 17 predefined keywords. Randomly selected text fragments were taken from the results for each keyword, amounting to 3771 text fragments from the records of 2832 patients.

**Outcome measures** We estimated precision, recall and F1 score for each NLP model. We examined sociodemographic and clinical variables in patients giving rise to the text data, and frequencies for each annotated violence characteristic.

**Results** Binary classification models were developed for six labels (violence presence, perpetrator, victim, domestic, physical and sexual). Among annotations affirmed for the presence of any violence, 78% (1724) referred to physical violence, 61% (1350) referred to patients as perpetrator and 33% (731) to domestic violence. NLP models' precision ranged from 89% (perpetrator) to 98% (sexual); recall ranged from 89% (victim, perpetrator) to 97% (sexual).

## Strengths and limitations of this study

- ► Previous natural language processing (NLP) models for extracting violence from mental health electronic health records (EHRs) have focused on single forms of violence, rather than capturing violence more broadly as this methodology does.
- ► This study fills a gap where newer fine-tuned transformer-based NLP models such as BioBERT have not yet been extensively researched in mental health applications.
- ► The methodology used can estimate the occurrence of clinical references to violence in EHRs but it cannot be used to estimate the prevalence of violent events without further assumptions.

**Conclusions** State of the art NLP models can extract and classify clinical text on violence from EHRs at acceptable levels of scale, efficiency and accuracy.

## INTRODUCTION

Interpersonal violence, defined as the intentional use of physical force or power, threatened or actual, against another person,[1] causes significant mental and physical morbidity.[2–4] Interpersonal violence may further be distinguished as domestic, physical and sexual violence. By its definition, interpersonal violence involves one or more perpetrator(s), one or more victim(s), and may also involve witnesses.

People with mental illness are more likely to experience violent victimisation compared with the general population.[5] For example, women with pre-existing mental illness are significantly more likely to experience victimisation compared with the general population with 15%–45% of patients reporting experiences of victimisation in the past year, and 40%–90% reporting lifetime victimisation.[5 6]

Domestic violence victimisation is also more frequently reported by people with mental illness, with 27% of women and 13% of men with severe mental illnesses (SMI) reporting experiences of domestic violence in the past year, compared with 9% and 5% respectively in general population samples.[6] Individuals with established mental disorders also experience greater occurrence of community and sexual violence compared with the general population.[7] Evidence also suggests associations between a diagnosis of SMI and higher perpetration of violence, compared with the general population.[8] Literature examining witnessing of violence is sparse, nevertheless there are some evidence indicating greater rates of witnessed violence among people with mental illness, and a detrimental impact of witnessed violence on mental health.[9][10] Adults who have experienced victimisation, and more specifically physical or sexual assault are at greater risk of mental disorders including post-traumatic stress disorder, depression and psychosis.[11][12] The impact of violence is multifold with significant economic, service and personal costs. For example, within mental health settings, violence occurs most frequently on inpatient psychiatric units, with an estimated cost of £20.5 million per year.[13]

Because of the consistent correlation between violence and psychiatric morbidity, mental health services are important settings for understanding and improving societal responses to violence. In particular, a large proportion of individuals in contact with mental health services have a history of violence exposure, including through victimisation, perpetration and/or witnessing interpersonal violence. Despite this, health services' data on interpersonal violence (including the role of patient as perpetrator, victim or witness, whether violence being referred to is domestic, physical and/or sexual) are inconsistent. Not all forms of violence are enquired about routinely by professionals, and some forms of violence are not routinely assigned diagnostic codes[14] and so are not easily identifiable in electronic records.

Electronic health records (EHRs) kept by mental health services offer a valuable resource to understand how and why interpersonal violence occurs in this population and examine how services respond to violence presentation (both as a victim and perpetrator) in relation to treatment and support. Improved understanding of interpersonal violence experienced by people using mental health services, and the response of professionals to violence, could improve care quality and patient safety. Mental health services can collect and record data on interpersonal violence, but structured data (eg, on violent incidents) are predominantly collected on individuals in inpatient settings, and not all forms of violence experienced and reported by patients may be recorded in this way.[15] We have previously employed text-processing rules to extract violence information from unstructured clinical text from EHRs, with a focus on physical violence.[16]

Natural language processing (NLP) methods offer a flexible automated approach to extracting text data from large bodies of unstructured text.[17][18] NLP models have been developed for the extraction of information on diagnosis, symptoms and treatment from clinical text.[15–17][19] Early NLP investigations using systems such as MedLEE relied on pattern matching and logical rules.[20] Developments in machine learning led to further advances and broad coverage applications, such as cTAKES (Clinical Text Analysis and Knowledge Extraction System)[21] and CLAMP (Clinical Language Annotation, Modeling, and Processing Toolkit).[22] In mental health research, NLP models have been developed to extract constructs such as phenotype mentions,[23] and symptoms of SMI.[22] Recent advances in deep neural network algorithms for NLP modelling, and particularly in transformer-based language models such as BERT[24] have shown promising results, also in the biomedical domain with, for example, the BioBERT model,[25] as shown in comparative analyses.[26][27] This research indicated that using a BioBERT fine-tuned algorithm outperformed most other algorithms. However, while these state-of-the-art approaches generate greater accuracy, they have not yet been extensively researched in mental health services. This paper does not evaluate or compare models but instead builds on these previous comparative analyses to evaluate the application of an existing model (BioBERT) onto a real-world problem.

Previous NLP approaches to capturing violence in mental health clinical text have had significant limitations such as focusing on single forms of violence, that is, physical assault victimisation,[28] rather than capturing a broad range of possibly co-existing violence characteristics (eg, perpetration, victimisation, witnessing), forms of harm (physical, sexual) and the nature of the relationship between victim and perpetrator (domestic, community) in the same NLP approach. Additionally, previous literature has used limited data, for example, only inpatient data. Unstructured data are challenging to examine due to complexity and volume, as evidenced by paucity of usage in previous studies.

## OBJECTIVE

To develop and evaluate NLP models for the extraction and classification of references to interpersonal violence from clinical text drawn from EHRs at a large mental health provider.

## METHODS
### Data source

Data were drawn from the Clinical Record Interactive Search (CRIS), a database of de-identified EHRs from the South London and Maudsley (SLaM) NHS Foundation Trust. SLaM provides specialist mental health services to around 1.3 million residents of four boroughs in South-East London (Lambeth, Southwark, Lewisham and Croydon). The CRIS database was developed in 2008 and allows researchers to access structured data (such as

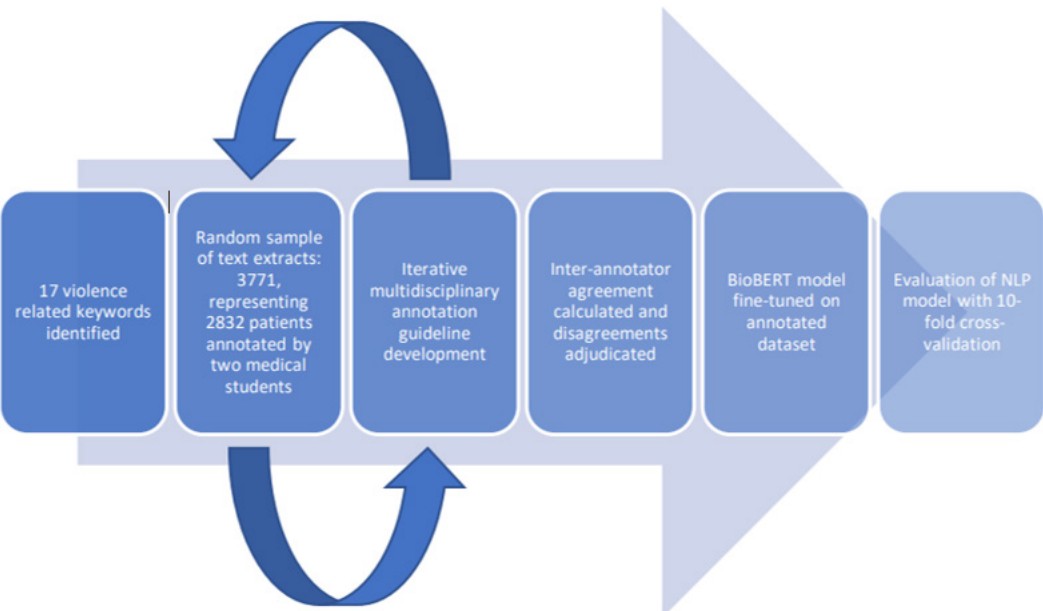

**Figure 1** Process of annotation, development and evaluation of natural language processing (NLP) models.

demographic data from forms) and unstructured data (such as free text entered by clinicians based on clinical encounters) for mental health research.[28–31] Currently, there are over 500 000 patient records represented in CRIS, CRIS, spanning 2007–present.

### Patient and public involvement statement

The CRIS database was developed with extensive service user involvement and adheres to strict governance frameworks managed by service users. Data are used in an entirely de-identified and secure format and all patients have the choice to opt-out of their de-identified data being used.[32] Each research project is reviewed by a service-user led oversight committee of the National Institute of Health Research Biomedical Research Centre.

### Methodological framework for annotation

Figure 1 summarises the annotation process. To generate annotation data for NLP model development, we generated a list of violence-related keywords based on the literature, clinical experience and informatics expertise (online supplemental appendix S1). An embedding model (Word2Vec trained on all CRIS records) was used to generate additional synonyms that were subsequently reviewed and included in the list of keywords. Furthermore, the embedding model was used to capture misspellings for each keyword.

For each keyword (17 in total; 7 nouns and 10 verbs), CRIS text containing that keyword was randomly sampled, extracting fragments containing the keyword and 300 text characters either side. We aimed to select 300 text fragments for each noun keyword, and 200 for each verb keyword. These text fragments were then annotated into labels for reference to violence by two clinical medical students (RB and MVW). Initial label definitions, results and queries were reviewed and discussed

by a multidisciplinary team which met weekly to formulate a set of further labels for NLP development based on the violence and mental health literature. Final labels are described further below. Weekly meetings also adjudicated on disagreements between annotators. We developed annotation guidelines (online supplemental appendix S2), which were iteratively developed based on discussion and queries raised by annotation. Discussions and rationale are detailed in online supplemental appendix S3. Interannotator agreement was estimated on a subset, using % agreement and Cohen's kappa.

### Labels used for annotating text fragments

Annotations were carried out for seven labels which were developed based on the WHO definition of interpersonal violence.[1] For each text label described below and exemplified in table 1, annotations classified text fragments as follows:

1. *Violence presence*: we annotated for the presence of any reference to *violence* in the text fragment, classifying fragments into: affirmed (where the characteristic was present), negated (characteristic absent) or irrelevant (where the keyword was employed in the text to refer to a context other than violence).
2. *Patient status*: we assigned three non-exclusive labels for the status of the patient within the text fragment, classified into whether the patient was perpetrator, victim and/or witness, to the text fragments where violence presence was annotated as affirmed. The label *perpetrator* was affirmed where the patient was referred to as the person using physical force or power, and the label *victim* was affirmed where the fragment referred to the patient as the person violence was used against. The label *witness* was affirmed where the fragment referred to the patient as having observed violence through seeing

**Table 1** Examples of text fragments, with keywords italicised, extracted for annotation in this study, alongside corresponding labels and assigned annotations

| Example of text fragment | Label | Annotation |
|---|---|---|
| 'They were *abused* in their childhood' | Violence presence, victim | Affirmed |
| 'Patient used to *hit* her partner' | Violence presence, perpetrator; physical, domestic | Affirmed |
| 'Patient *stabbed* his roommate' | Violence presence, perpetrator; physical, domestic | Affirmed |
| 'Expressed a lot of interest in *violence*, nazism' | Violence presence | Irrelevant |
| 'No *violence* or aggression noted' | Violence presence | Negated |

or hearing violence occurring and this was the primary description, that is, they were not a victim or perpetrator of that violence.

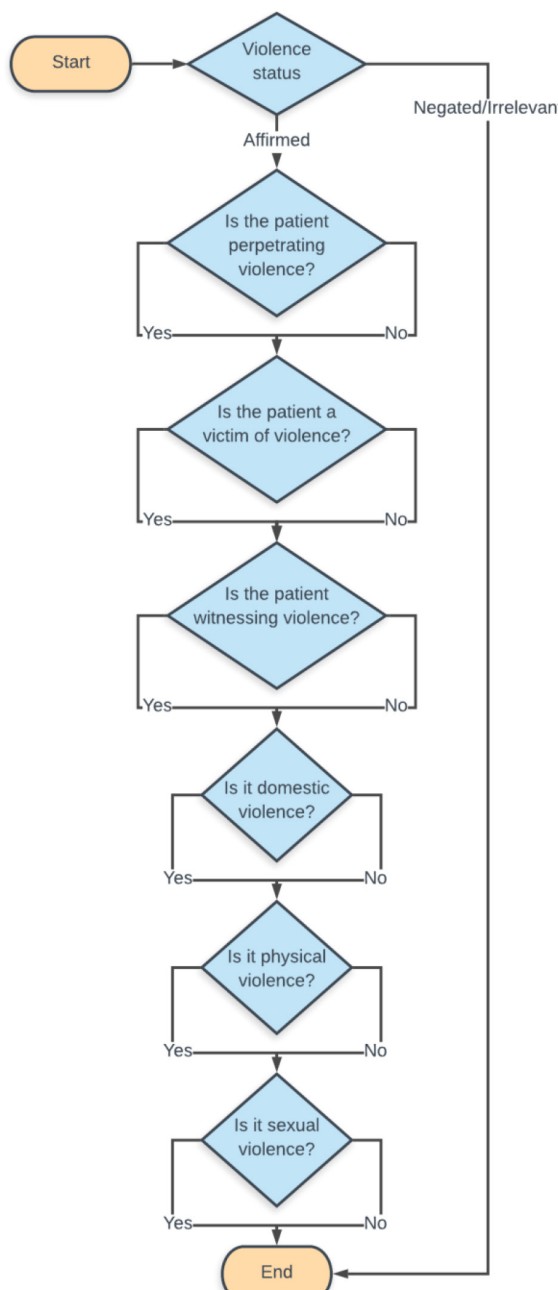

**Figure 2** Flow chart of extract annotation process.

3. *Violence type*: we also assigned labels classifying violence by forms of harm (physical, sexual), and based on the relationship between the victim and perpetrator (domestic). The label *physical* was affirmed where the text fragment referred to violence which used physical force, or resulted in or had a high likelihood of resulting in physical injury. The label *sexual* was affirmed where the text fragment referring to violence included unwanted sexual acts, unwanted sexual comments or advances or unwanted attempts to obtain a sexual act. This included references to rape, sexual harassment, sexual assault, forced marriage, stalking and reproductive coercion and control. The label *domestic* was affirmed where text referred to violence between family members, intimate partners, ex-intimate partners and household members. As with *patient status*, these labels were also non-exclusive, so that text fragments could include any combination of affirmed labels listed above.

The labels listed above were annotated in the following steps. First, each text extract was annotated for the violence presence label, classifying the fragment of text as affirmed, negated or irrelevant. If this label was affirmed, the fragment was further annotated with labels for patient status (victim, perpetrator and/or witness) and violence type (domestic, physical, sexual). This is represented in figure 2.

## NLP model development

For the development of NLP models, we used the pretrained BioBERT model[25] and fine-tuned it on the annotated dataset. Each set was generated independently (ensuring no overlap). Three datasets were used for: model testing and training (development stage, 3771 sentences) model fine-tuning (1411 sentences), and model blind testing (100 sentences). We aimed to produce seven binary classification models for each annotated label. We evaluated the models with 10-fold cross-validation, comprising 10% annotated text extracts for testing, and 90% text extracts in training, in each fold. We estimated standard markers of NLP performance: precision (or positive predictive value), recall (or sensitivity) and F1 score (the harmonic mean of precision and recall), using weighted averages to take into account the dataset's imbalance (ie, differing numbers of extracts generated for each keyword). Reported scores corresponded to the

**Table 3** Proportion of each label in the training and testing dataset—affirmed or negated/irrelevant

| Annotation label | Affirmed, N (%) | Negated or irrelevant, N (%) | Total |
|---|---|---|---|
| Violence presence | 2199 (58) | 1572 (42) | 3771 |
| Patient status: perpetrator | 1350 (61) | 849 (39) | 2199 |
| Patient status: victim | 731 (33) | 1468 (67) | 2199 |
| Violence type: domestic | 723 (33) | 1476 (67) | 2199 |
| Violence type: physical | 1724 (78) | 475 (22) | 2199 |
| Violence type: sexual | 353 (16) | 1846 (84) | 2199 |

Each text extract was first annotated for the violence presence label, then if this was affirmed, further annotated for the other labels related to patient status and violence type (see figure 1 for further details). Therefore, denominator totals for the violence presence label is larger than that for the other labels.

mean across the 10 test sets. The model fine-tuning test set was run on 1411 sentences extracted randomly from the CRIS database (sentences not used to train the model or test the development). This set was randomly sampled using the clinical records not previously used for training, and the same list of 27 keywords. This set was then manually annotated following the same guidelines as the training set and compared against the output generated by the NLP model. Weighted loss functions (cross entropy loss with custom weight parameters) were used to account for unbalanced datasets. The blind model testing set of 100 sentences were generated to review sample generation, annotation and model assessment.

For descriptive purposes, we examined sociodemographic and diagnostic characteristics of patients whose records gave rise to the text extracts used for NLP development, and also assessed two-way overlap of affirmed non-exclusive annotation labels.

## RESULTS

Sampling of text for NLP development resulted in 14 of the 17 keywords being sampled as planned. Three keywords generated a smaller number of selected fragments: 'rape' 188; 'fought' 124 and 'strangul' 59, resulting in a final annotation dataset of 3771 text extracts. Fine-tuning of the pretrained BioBERT model using the annotation dataset resulted in six binary classification models—for one annotation label (witness) we were unable to generate a model due to insufficient data size (n=53). The performance of

each of the six NLP models is reported in table 2. Two types of testing were carried out to evaluate the model's suitability. A 10-fold cross-validation was conducted on the training and testing dataset (comprising 3771 text extracts), for which the average performance on the test sets is reported. Precision ranged from 89% (for the perpetrator label) to 98% (sexual) and recall ranged from 89% (victim, perpetrator) to 97% (sexual). Inter-annotator agreement was high: 82%–96% (60%–85% Cohen's kappa) for the six annotation labels. Additionally, a separate blind test consisting of 100 newly annotated sentences not used for model training or fine-tuning was assessed. For this blind test, we used a confidence level of 90% (meaning only sentences that the model classified with 90% or above confidence were included, in order to eliminate 'confusing statements', which made up 1% of the dataset). This 90% threshold was then used for application deployment.

Table 3 shows the breakdown of annotation labels across the training and testing dataset at an annotation level. Overall, 58% (2199) of the text extracts were affirmed for violence presence. Of these, the proportion of affirmed examples for patient status ranged from 33% (victim) to 61% (perpetrator), and for violence type from 16% (sexual) to 78% (physical). Some affirmed labels overlapped, because a fragment of text could have contained multiple references to different types of violence (eg, both physical and sexual violence) and different patient statuses (eg, as both perpetrator and victim). Table 4

**Table 2** NLP model performances on the training and testing dataset (3771 text extracts) and well as a blind test set with a 90% probability threshold (100 sentences) for the six labels

| Annotation label | Training set (average score on 10-fold cross-validation) | | | Blind test set |
|---|---|---|---|---|
| | Precision | Recall | F1 score | F1 score |
| Violence presence | 93% | 93% | 93% | 95% |
| Patient status: perpetrator | 89% | 89% | 89% | 85% |
| Patient status: victim | 91% | 89% | 91% | 90% |
| Violence type: domestic | 94% | 94% | 94% | 93% |
| Violence type: physical | 91% | 92% | 91% | 98% |
| Violence type: sexual | 98% | 97% | 97% | 93% |

**Table 4** Overlap of labels present in affirmed annotations, showing the number and percentage of annotations that shared different labels

| | Perpetrator, N (%) | Victim, N (%) | Sexual, N (%) | Physical, N (%) |
|---|---|---|---|---|
| Perpetrator | – | – | – | – |
| Victim | 113 (8.4) | – | – | – |
| Sexual | 150 (11.1) | 199 (27.2) | – | – |
| Physical | 1078 (79.9) | 616 (84.3) | 304 (86.1) | – |
| Domestic | 331 (24.5) | 318 (43.5) | 104 (29.5) | 593 (34.4) |
| *Column total* | *1350 (100.0)* | *731 (100.0)* | *353 (100.0)* | *1724 (100.0)* |

describes these overlaps. For instance, the number of examples where *perpetrator* was affirmed (n=1350) overlapped with affirmed for *physical* violence in almost 80% of the examples (n=1078), while the overlap with *victim* was rarer (8%, n=113).

Table 5 reports model-to-annotator agreement, that is, agreement between the two annotators combined and the model. Table 6 reports errors made by the model. Total errors represent instances where the model predicted differently to the annotators, while 'false positives' represent instances where the model classified an instance as violent while annotators classified it as irrelevant or negated. 'False negatives' are instances that the model classified as irrelevant or negated while annotators markers as affirmed. Both tables were computed using the model training and testing set of 3771 sentences and indicate high agreement between the models and annotators.

Demographic features of the patients included in the training and testing dataset sample are presented in table 7. The sample represented in total 2832 patients, of whom 57% were female, 45% were aged 40–60 years at the time the fragment was extracted, 66% were single, 52% were of white ethnic background and 36% were diagnosed with psychotic disorders (International Classification of Diseases-10 codes F20–29).[33] This NLP approach captures references to violence occurring in a large body of patients presenting to a mental health service who are not necessarily presenting for violence-related reasons.

**Table 5** Kappa agreement between manually and automatically assigned categories in the training and testing set (3771 sentences)

| Annotation label | Model-to-annotator agreement |
|---|---|
| Violence presence | 98.1% |
| Patient status: perpetrator | 97.4% |
| Patient status: victim | 96.2% |
| Violence type: domestic | 98.7% |
| Violence type: physical | 98.3% |
| Violence type: sexual | 96.8% |

## DISCUSSION

To our knowledge, this is the first study that has used a NLP approach to code free-text data from a large and diverse source of electronic mental health records to ascertain violence according to presence, agent and type. There is limited previous research examining NLP to extract violence-related information in EHRs but includes using text rules to ascertain violent behaviour as antecedents to supervised confinement[16] and the employment of a bag of words machine learning approach to extract information on physical assault victimisation in CRIS data.[28] Our approach captures a much broader range of experiences. We successfully developed an annotated dataset of clinical text references to interpersonal violence, using a multidisciplinary clinical academic group. We used this dataset to develop binary classification NLP models for extracting and classifying clinical text fragments referring to interpersonal violence in mental health EHRs, including for patient status (perpetrator, victim) and violence type (domestic, physical, sexual). Models were developed with a state-of-the-art NLP algorithm (fine-tuned BioBERT) and displayed very good performance based on accepted evaluation criteria. A planned NLP model for extracting references to witnessed violence was not successfully developed due to an insufficient sample size.

This study had several limitations that need to be borne in mind when interpreting the findings. Although agreement between annotators was generally good, some disagreements occurred. Examples of disagreements included ascertaining the status of the patient as perpetrator/victim/witness in a text fragment and whether 'fighting' was considered interpersonal violence or a colloquial term for verbal arguments. As our focus was on interpersonal violence, we did not annotate or develop models for individuals forced to fight in armed conflict. Keywords used in this study were selected to capture as many instances of interpersonal violence in clinical text as possible but may not have captured all violence categories of 'hidden violence'. For instance, keywords related to female genital mutilation, forced marriage, trafficking, neglect, sensory deprivation, harrassment, stalking or reproductive coercion and control were not included, and would need separate analysis for NLP development.

**Table 6** Model error analysis on training and testing set (3771 sentences)

| Annotation label | False positives | False negatives | Total number of errors |
|---|---|---|---|
| Violence presence | 24 | 10 | 34 |
| Patient status: perpetrator | 35 | 12 | 47 |
| Patient status: victim | 5 | 40 | 45 |
| Violence type: domestic | 10 | 5 | 15 |
| Violence type: physical | 7 | 27 | 34 |
| Violence type: sexual | 11 | 10 | 21 |

These specific forms of violence could, however, be readily addressed using an 'add-on' to the approach presented here. Similarly, emotional and psychological violence were not included in the list of keywords, and capturing these characteristics through NLP is likely to be more complex because of the broader way in which this is likely to be described in clinical text. Annotation was carried out on the basis of the meaning and sense of the text fragments sample in this study, rather than entire EHR documents. It is possible that annotations based on entire documents would have delivered slightly different results, but this would have been challenging to implement given the quantity of documents that would be needed to be manually labelled in order to capture enough examples. This model could potentially be improved by additional fine-tuning on a clinical dataset. Lastly, we present an NLP approach to extracting clinical text fragments which refer to violence from mental health records. Given the sufficiently accurate performance reported in this study, this approach can extract references to violence where it is written down in clinical records, but is restricted to these recorded instances, and cannot be considered a method for measuring prevalence of all experienced violence (without further assumptions which are likely to depend on the situation). This is likely to continue to require asking patients themselves, or linking data from other sources such as hospital data or crime records. Indeed, these approaches could be helpful in understanding the processes by which violence is identified by clinicians and recorded in clinical notes.

This study had some specific advantages. Annotations were derived from rich and diverse free-text data from service-users' clinical notes, as opposed to structured data. The data are derived from progress notes entered by wide array of clinical groups and professionals, therefore increasing the chances of detecting violence information. Furthermore, given the longitudinal nature of our dataset, information was recorded over a 16-year period (2007–2019), which further increases our ability to detect experiences of violence, if recorded in the clinical notes. All annotations were coded using human annotators, with high interannotator agreement, which support the robustness of our approach. Lastly, we used a state-of-the-art NLP pretrained transformer model, BioBERT, which has been shown to outperform methods more traditionally used for symptoms detection such as support vector machine[26] and this allowed us to develop fine-tuned models with very promising results. As this study did not make comparative evaluations, it is possible that a simpler baseline model could also provide a similar level of performance, while requiring fewer resources. The BioBERT model is readily deployable and interpretable, with all scripts made publicly available on GitHub. The fine-tuned model is relatively light (450 Mb) and easy to deploy in clinical settings. The model was run in this study using Graphics Processing Unit-accelerated analytics, which may limit replicability.

NLP offers a reliable, automated, scalable approach to extracting summary information from EHRs. Reasonably accurate methods for extracting clinical text referring to interpersonal violence can support further research on correlates of clinically identified violence, and how professionals and services respond to violence. Evaluation of NLP-derived violence indicators in relation to linked external data on hospital admissions, GP registers and crimes data could allow some assessment of the reliability with which clinicians record violence when it occurs. Based on the interoperability of NLP algorithms across different EHR formats, models developed in this study could be feasibly applied to mental health EHRs in other sites (including CRIS databases elsewhere in the UK) and other free-text containing EHRs. The main limitation to porting this model to other sites would be differences in language that may lead to a slightly different list of keywords. However, the current list already captures many common terms related to violence and was reviewed by clinicians that work in several NHS Trusts to ensure generalisability. Furthermore, we have developed an NLP tool that is easily adaptable and allows quick fine-tuning and deployment of the model if needed.

There is also a significant clinical need for methods which can accurately summarise relevant historic information for clinicians to review and use in decision making at the point of care. This is particularly relevant in the arena of violence reduction and mental health, where serious incident reviews and charitable organisations have called for renewed attention to the need for accurate summaries of previous violence, and improved information sharing with other agencies.[34] By summarising previous exposure to violence, clinical information systems might improve

**Table 7** Characteristics of patients whose text extracts were annotated as part of this study

| | Frequency, N (%) |
|---|---|
| Age (years) | |
| <20 | 167 (5.9) |
| 20 to <40 | 791 (27.9) |
| 40 to <60 | 1273 (45.0) |
| 60 to <80 | 458 (16.2) |
| 80< | 141 (5.0) |
| Missing | 2 (0.1) |
| Gender | |
| Male | 1216 (42.9) |
| Female | 1614 (57.0) |
| Missing | 2 (0.1) |
| Marital status | |
| Single | 1865 (65.9) |
| Married/Cohabiting | 344 (12.2) |
| Divorced/Separated | 262 (9.3) |
| Widowed | 85 (3.0) |
| Missing | 276 (9.8) |
| Ethnicity | |
| White | 1482 (52.3) |
| Black | 104 (3.7) |
| Asian | 160 (5.7) |
| Mixed | 885 (31.3) |
| Other | 90 (3.2) |
| Missing | 111 (3.9) |
| ICD-10 diagnosis | |
| F0–9: organic, including symptomatic, mental disorders | 185 (6.5) |
| F10–19: mental and behavioural disorders due to psychoactive substance use | 94 (3.3) |
| F20–29: schizophrenia, schizotypal and delusional disorders | 1031 (36.4) |
| F30–39: mood (affective) disorders | 451 (15.9) |
| F40–49: neurotic, stress-related and somatoform disorders | 203 (7.2) |
| F50–59: behavioural syndromes associated with physiological disturbances and physical factors | 18 (0.6) |
| F60–69: disorders of adult personality and behaviour | 236 (8.3) |
| F70–79: mental retardation | 53 (1.9) |
| F80–89: disorders of psychological development | 98 (3.5) |
| F90–99: behavioural and emotional disorders with onset usually occurring in childhood and adolescence and unspecified mental disorder | 211 (7.6) |

Continued

**Table 7** Continued

| | Frequency, N (%) |
|---|---|
| No axis 1 diagnosis | 25 (0.9) |
| G: diseases of the nervous system, X: intentional self-harm, assault or Z: factors influencing health status and contact with health services | 163 (5.8) |
| Missing | 64 (2.3) |
| Total | **2832** (100.0) |

*ICD-10 categories G and X were combined with ICD-10 category Z due to small numbers of participants (n<10) in these categories, in order to limit identification of participants.
ICD, International Classification of Diseases.

the efficiency of clinical encounters, for example, by reducing clinical time taken up with collecting information on previous violence. Future research might also consider the clinical benefits of accurate summaries of previous interpersonal violence which might be reported to treating clinicians in real time, to aid decision making. We suggest that attention is warranted into the ethical and regulatory challenges of translating NLP methods for violence into practice.

We were unable to model witnessing violence using NLP in this study. This could form the focus of future research based on the possible impact of witnessing violence on health.[35] The infrequency of witnessed violence during the annotation process may reflect that clinicians are not frequently enquiring about these experiences, or they are not being recorded.

We did not consider temporality of violence mentions, but this is something that we hope to integrate in future to improve the usability of the models.

**CONCLUSION**

We have demonstrated that it is possible to use state-of-the-art NLP methods to extract clinical text referring to violence (including distinguishing patients as perpetrators and victims, as well as violence types such as physical, domestic and sexual) at scale with acceptable accuracy in mental health EHRs. This could support further research into the pathways by which violence is identified in clinical practice, and the effectiveness of systems for identifying, assessing and managing interpersonal violence. However, while NLP approaches might offer a sufficiently accurate method for summarising violence-related information to aid clinical decision making, considerable ethical and regulatory questions remain.

**Author affiliations**
[1]School of Medical Education, Guy's, King's and St Thomas' School of Medicine, London, UK
[2]Section of Women's Mental Health, Department of Health Services and Population Research, King's College London, London, UK

³Psychological Medicine, Institute of Psychiatry, Psychology and Neuroscience, King's College London, London, UK
⁴Biostatistics and Health Informatics, King's College London, London, UK
⁵Health Data Research UK, London, UK
⁶South London and Maudsley Mental Health NHS Trust, London, UK

**Contributors** RB drafted the first version of the manuscript. RB and MVW annotated the dataset. AM developed the NLP models. VB, GK-S and RS provided clinical and epidemiological input in relation to the motivation for the work, annotation guideline development and definitions. SV and AR provided technical input and guidance in relation to the NLP model development. All authors drafted and/or critically revised the manuscript and approved the final version of the manuscript. RS is the guarantor for this study.

**Funding** This project received no specific funding. The Clinical Record Interactive Search (CRIS) system was funded and developed by the National Institute for Health Research (NIHR) Biomedical Research Centre at South London and Maudsley NHS Foundation Trust and King's College London and by a joint infrastructure grant from Guy's and St Thomas' Charity and the Maudsley Charity (grant number BRC-2011-10035). RS, SV, AR and GK-S receive salary support from the NIHR Biomedical Research Centre at South London and Maudsley NHS Foundation Trust and King's College London. RS is a NIHR Senior Investigator. GK-S has received support from the Early Career Research Award, funded by the NIHR Maudsley BRC. VB receives salary support from King's College London via a secondment to Lambeth Council. AR receives support from Health Data Research UK, an initiative funded by UK Research and Innovation, Department of Health and Social Care (England) and the devolved administrations, and leading medical research charities. The views expressed are those of the author(s) and not necessarily those of the NHS, the NIHR or the Department of Health.

**Disclaimer** The funders had no role in the study design; in the collection, analysis and interpretation of the data; in the writing of the report and in the decision to submit the paper for publication.

**Competing interests** RS has received research support in the last 36 months from Janssen, Takeda and GSK. GK-S has received funding from Janssen and Lundbeck. SV has received funding from Janssen. AM has received funding from Takeda California.

**Patient consent for publication** Not applicable.

**Ethics approval** The CRIS database has received ethical approval for secondary analysis (Oxford Research Ethics Committee, reference 18/SC/0372).

**Provenance and peer review** Not commissioned; externally peer reviewed.

**Data availability statement** Data are available on reasonable request. The CRIS database has received ethical approval for secondary analysis: Oxford REC C, reference 18/SC/0372. On request, and after appropriate arrangements, the data and modelling employed in this study can be viewed within the secure system firewall.

**ORCID iDs**
Riley Botelle http://orcid.org/0000-0002-3052-5698
Giouliana Kadra-Scalzo http://orcid.org/0000-0003-3182-905X

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
