## [Reviewer comments · BMJ Open]

ARTICLE DETAILS

TITLE (PROVISIONAL)	Can natural language processing models extract and classify instances of interpersonal violence in mental healthcare electronic records: an applied evaluative study
AUTHORS	Botelle, Riley; Bhavsar, Vishal; Kadra-Scalzo, Giouliana; Mascio, Aurelie; Williams, Marcus; Roberts, Angus; Velupillai, Sumithra; Stewart, Robert

VERSION 1 – REVIEW

REVIEWER	Enrique Baca-Garcia Universidad Autónoma de Madrid
REVIEW RETURNED	26-Jun-2021

GENERAL COMMENTS	The article is very well written. It is very interesting in its approach and very promising, technically it is impeccable. The title should include "interpersonal" violence. My main concern is the definition of violence, which is too narrow. For example, it leaves out bullying in the school and work environment. One of the strengths of this methodology is to uncover unrecorded violence. This approach is an opportunity especially for women and young adults. That is why it is important that the authors broaden the definition of interpersonal violence and include the categories identified by WHO: "Interpersonal violence refers to violence between individuals, and is subdivided into family and intimate partner violence and community violence. The former category includes child maltreatment; intimate partner violence; and elder abuse, while the latter is broken down into acquaintance and stranger violence and includes youth violence; assault by strangers; violence related to property crimes; and violence in workplaces and other institutions." https://www.who.int/violenceprevention/approach/definition/en/
--

REVIEWER	Trevor Cohen University of Washington, Department of Biomedical Informatics and Medical Education
REVIEW RETURNED	14-Jul-2021

GENERAL COMMENTS	This paper describes the development and evaluation of a natural language processing (NLP) model to identify text describing violent incidents in clinical narratives. The strengths of the paper include the societal importance of the problem, the multidisciplinary approach to developing a coding framework, the granularity of the resulting categories, and the impressive performance obtained with contemporary transfer learning approaches. These strengths notwithstanding, the paper leaves some important questions unanswered on account of the use of a single NLP approach without
---

	any points of comparison. In particular, readers with limited access to the computational resources (GPUs) required to train and run BERT-based models may wish to know if a model with this many parameters is really required to solve the problem at hand. If similar results were obtained with a simpler model, this would provide potential for broader deployment. In any event baseline performance estimates for relatively straightforward (and readily deployable and interpretable) models would be helpful to provide an assessment of the difficulty of the problem concerned. Standard baselines might include a logistic regression model or support vector machine trained on bag-of-words features, for example, or even a rule-based model although this might be difficult to develop without bias given the use of representative keywords to construct the corpus. This use of violence-related keywords to develop a training and evaluation set seems as though it has some limitations beyond those addressed in the paper. For example, is it possible that models may then perform better in contexts in which violent incidents are described using terms in this set of 27 than when other terms are used to describe similar incidents? In theory, a pre-trained transformer should be able to generalize to identify the same event expressed in different words. However, evaluation in the context of a set defined by these keywords would not permit characterization of the extent to which this really occurs. One way around this problem might involve using a measure of term similarity (e.g. between word embeddings) to extend the set of terms that were manually defined, or extend the constraints for the construction of development and/or test sets to include their semantic relatives. The performance on the 1411 randomly selected sentences may also provide some indication – assuming these sentences were selected without recourse to the keywords. However, this is something of a Catch-22 as it seems as though the base rate for description of violent incidents amongst randomly selected sentences would be very low. Further details on this test set (i.e. how it was selected and annotated) would be very helpful here. There are also some apparent inconsistencies in the way in which this set is described – we have 1411 randomly selected sentences in the methods, 1000 newly-annotated sentences in the results, and a blind test set of 100 sentences above a 90% probability threshold in Table 2. Are these sets related to one another somehow? Regarding the configuration of BERT, did the authors consider a single multi-label model as an alternative to training seven independent models? Aside from the advantages with respect to GPU memory and such, BERT may be able to leverage information acquired while learning to assign one label when attempting to assign others. Another architectural consideration that seems worth considering involves the use of weighted loss functions to account for class imbalance, although the motivation for further methodological development may be limited given the levels of performance already attained. Another methodological question concerns the choice of BERT model. While BioBERT seems like a reasonable choice of model, it is trained on the biomedical literature – one wonders how much the additional information obtained in this way is informative with respect to descriptions of violence, and it would be interesting to know
--	--

	whether or not there are performance advantages relative to the un-augmented BERT model. The high inter-annotator agreement exhibited on this task may provide a ceiling for NLP model performance. Thus, it would be informative to know the average model-to-annotator agreement. It may be the case that the authors approach has already achieved human performance, and in any event it would be good to know how close it is to a reasonable ceiling. One way to assess this might be to recalculate Kappa using automatically assigned categories, as compared with the categories assigned by each rater. Though the claims in the paper are generally well supported by the evidence that accompanies them, one wonders about the assertion that models developed in the current study could be applied at other sites. Do the authors have any concerns about the portability of this model (given that this is a known limitation of clinical NLP models in general), and (if so) any suggestions as to how these concerns might be mitigated at the point of redeployment? The authors also raise the issue that annotation at document level may have delivered different results. This seems a valid concern – for one thing, documents are likely to extend beyond the window of tokens that BioBERT permits as input. However, annotating at fragment level seems appropriate here. A broader concern is whether or not the measures of performance provided accurately reflect how the model might perform in the context of notes that have not been segmented or enriched for terms indicative of violence. Some automated method of segmentation would be needed, and even small decrements in precision may result in retrieval of a large number of irrelevant sentences given that the sentence-level prevalence of violent incidents in the wild is likely to be much lower than their prevalence in the set the authors have curated. It would strengthen the paper to add some discussion of factors that may influence performance in practice, and the implications of resulting changes in performance characteristics for the intended applications of the model. In summary, this is a well-constructed paper that documents very promising performance in identifying mentions of violent incidents in clinical notes, obtained by fine-tuning the BioBERT model. The paper would be strengthened by including a simple baseline model, so readers can determine whether or not using a model the size of BioBERT is justified. From a methodological perspective there are a number of architectural variants (e.g. multilabel configuration, weighted loss, BERT sans Bio) that may be worth considering, though this may be beyond the scope of the current paper, and it seems likely that performance is already close to a ceiling. This issue aside, there are some details of the evaluation that require clarification in order to assess the extent to which performance may be artificially elevated by limiting training, validation and perhaps test sets to those sentences containing a small set of manually defined keywords. - Is 'strangul' in the Results section really one of the keywords that was used? It seems an odd choice.
--	---

VERSION 1 – AUTHOR RESPONSE

Reviewer: 1

Dr. Enrique Baca-Garcia, Universidad Autónoma de Madrid

Reference	Feedback	Action taken
1	The title should include "interpersonal" violence.	The title has been changed to "Can natural language processing models extract and classify instances of interpersonal violence in mental healthcare electronic records: an applied evaluative study".
2	My main concern is the definition of violence, which is too narrow. For example, it leaves out bullying in the school and work environment . . . that is why it is important that the authors broaden the definition of interpersonal violence and include the categories identified by WHO	We appreciate this concern. Our definition of violence utilised the WHO categories of interpersonal violence but was not sufficient to capture all aspects of violence, and this is considered in the discussion. It would require significant further work to reconceptualise violence, and whilst we think this work would be extremely valuable, it is better suited to further development following on from this paper. Expansions of this model could cover more diverse forms of violence, or focus on specific kinds.

Reviewer: 2

Dr. Trevor Cohen, University of Washington

Referen ce	Feedback	Action taken
3	The paper leaves some important questions unanswered on account of the use of a single NLP approach without any points of comparison . . . readers with limited access to the computational resources (GPUs) required to train and run BERT-based models may wish to know if a model with this many	Much of the NLP work underlying that research has previously been covered and we are sorry that this was not sufficiently clear in the originally submitted manuscript. We have previously run an extensive comparison of various NLP approaches for text classification using several biomedical shared tasks (https://aclanthology.org/2020.bionlp-1.9/, manuscript reference 26), and this work was subsequently reproduced on a CRIS internal dataset (http://healtex.org/healtac-2020/programme/). This is now more clearly referenced and discussed in the introduction. This previous research indicated that using a BioBERT fine-tuned algorithm led to the best ratio of text pre-processing/performance. The BioBERT model is readily deployable and interpretable, all scripts have been made publicly available on GitHub. (https://github.com/KCLaurelie/prometheus/tree/master/sentence_classifier/sentence_classifier) This is now referenced in the Discussion section.

	parameters is really required to solve the problem at hand. If similar results were obtained with a simpler model, this would provide potential for broader deployment. In any event baseline performance estimates for relatively straightforward (and readily deployable and interpretable) models would be helpful to provide an assessment of the difficulty of the problem concerned. Standard baselines might include a logistic regression model or support vector machine trained on bag-of-words features, for example, or even a rule-based model although this might be difficult to develop without bias given the use of representative	
--	---	--

	keywords to construct the corpus.	
4	This use of violence-related keywords to develop a training and evaluation set seems as though it has some limitations beyond those addressed in the paper. For example, is it possible that models may then perform better in contexts in which violent incidents are described using terms in this set of 27 than when other terms are used to describe similar incidents? In theory, a pre-trained transformer should be able to generalize to identify the same event expressed in different words. However, evaluation in the context of a set defined by these keywords	We apologise that this was not sufficiently clear in the manuscript. We have used an embedding model (Word2Vec trained on all CRIS records) to generate additional synonyms, that have subsequently been reviewed by clinicians and are included in the list of 17 keywords used. Furthermore, the embedding model was used to capture misspellings for each keyword (i.e. sentences are generated based on the original list of 17 keywords + their respective synonyms). This is now referenced in the methodology.

	would not permit characterization of the extent to which this really occurs. One way around this problem might involve using a measure of term similarity (e.g. between word embeddings) to extend the set of terms that were manually defined, or extend the constraints for the construction of development and/or test sets to include their semantic relatives.	
5	The performance on the 1411 randomly selected sentences may also provide some indication – assuming these sentences were selected without recourse to the keywords. However, this is something of a Catch-22 as it seems as	This separate blind test set was randomly sampled using the clinical records not previously used for training/initial testing, and the same list of 17 keywords. This set was then manually annotated following the same guidelines as the training set and compared against the output generated by the NLP model. This is now referenced under “NLP model development”.

	though the base rate for description of violent incidents amongst randomly selected sentences would be very low. Further details on this test set (i.e. how it was selected and annotated) would be very helpful here.	
6	There are also some apparent inconsistencies in the way in which this set is described – we have 1411 randomly selected sentences in the methods, 1000 newly-annotated sentences in the results, and a blind test set of 100 sentences above a 90% probability threshold in Table 2. Are these sets related to one another somehow?	Each set was generated independently (making sure of no overlap) and doubly annotated. Several datasets were used for: model fine-tuning, model initial testing (development stage), model blind testing. This is now referenced under “NLP model development”.
7	Regarding the configuration of BERT, did the authors	We tried a single multi-label modelling approach, however the overlap of several labels (e.g. a violent event can be both classified as domestic and sexual) led to a list of complex annotations and loss in accuracy. On top of improving accuracy, having independent models for each type of violence

	consider a single multi-label model as an alternative to training seven independent models? Aside from the advantages with respect to GPU memory and such, BERT may be able to leverage information acquired while learning to assign one label when attempting to assign others.	facilitates extracting results for researchers, who are typically interested in one specific type of violence (or in combining two specific types).
8	Another architectural consideration that seems worth considering involves the use of weighted loss functions to account for class imbalance, although the motivation for further methodological development may be limited given the levels of performance already attained.	We used weighted loss functions (cross entropy loss with custom weight parameters) to account for unbalanced datasets. This is now referenced under "NLP model development".
9	The high inter-annotator	The model's performance (F1, precision, recall) and newly calculated kappa between manually and automatically assigned categories indicate a

	agreement exhibited on this task may provide a ceiling for NLP model performance. Thus, it would be informative to know the average model-to-annotator agreement. It may be the case that the authors approach has already achieved human performance, and in any event it would be good to know how close it is to a reasonable ceiling. One way to assess this might be to recalculate Kappa using automatically assigned categories, as compared with the categories assigned by each rater.	high model-to-annotator agreement. These have been added to the paper as Table 6 and Table 7. As suggested, we recalculated Kappa between manually and automatically assigned categories (“model-to-annotator agreement”), using training + testing sets:  - 98.1% for status - 96.2% for victim - 97.4% for perpetrator - 98.7% for domestic - 98.3% for physical - 96.8% for sexual Model error analysis on training+testing set: 196 errors  - 34 for status (24 FP - classed as 1 that should have been 0) - 45 for victim (5 FP) - 47 for perpetrator (35 FP) - 15 for domestic (10 FP) - 34 for physical (7 FP) - 21 for sexual (11 FP)
10	Though the claims in the paper are generally well supported by the evidence that accompanies them, one wonders	The main limitation to porting this model to other sites would be differences in language that may lead to a slightly different list of keywords. However, the current list already captures many common terms related to violence and was reviewed by clinicians that work in several Trusts to ensure generalisability. Furthermore, we believe we have developed an NLP tool that should be easily adaptable and allows quick re fine-tuning/deployment of the model if needed. This is now included in the Discussion section.

	about the assertion that models developed in the current study could be applied at other sites. Do the authors have any concerns about the portability of this model (given that this is a known limitation of clinical NLP models in general), and (if so) any suggestions as to how these concerns might be mitigated at the point of redeployment ?	
11	The authors also raise the issue that annotation at document level may have delivered different results. This seems a valid concern – for one thing, documents are likely to extend beyond the window of tokens that BioBERT	The sentences are extracted from documents using the pre-defined list of keywords, as a consequence any portion of text that does not contain relevant keyword would be excluded. We do accept the Reviewer’s point here; however, document level annotation is challenging to implement given the quantity of documents needed to manually label in order to capture enough examples. This is now expanded upon in the Discussion.

	permits as input. However, annotating at fragment level seems appropriate here. A broader concern is whether or not the measures of performance provided accurately reflect how the model might perform in the context of notes that have not been segmented or enriched for terms indicative of violence. Some automated method of segmentation would be needed, and even small decrements in precision may result in retrieval of a large number of irrelevant sentences given that the sentence-level prevalence of violent incidents in the wild is likely to be much lower than their prevalence in the set the	
--	---	--

	authors have curated. It would strengthen the paper to add some discussion of factors that may influence performance in practice, and the implications of resulting changes in performance characteristics for the intended applications of the model.	
12	The paper would be strengthened by including a simple baseline model, so readers can determine whether or not using a model the size of BioBERT is justified. From a methodological perspective there are a number of architectural variants (e.g. multilabel configuration, weighted loss, BERT sans Bio) that may be worth considering, though this may be	We agree with this point. Other models were tested as part of previous comparative analyses referenced in the paper and BioBERT showed significant improvement compared to simpler architectures such as Support Vector Machine. BioBERT fine-tuning does not need significant computer resources (as opposed to re-training from scratch a transformer-based model) and only takes a couple of hours to complete when running 10-fold cross validation. Fine-tuning one model takes less than an hour. Finally, the fine-tuned model is relatively light (450Mb) and easy to deploy in clinical settings. Regarding the evaluation, the use of a blind test dataset confirmed the good performance of the model on previously unseen data.

	beyond the scope of the current paper, and it seems likely that performance is already close to a ceiling. This issue aside, there are some details of the evaluation that require clarification in order to assess the extent to which performance may be artificially elevated by limiting training, validation and perhaps test sets to those sentences containing a small set of manually defined keywords.	
13	Is 'strangul' in the Results section really one of the keywords that was used? It seems an odd choice.	This is to capture "strangulation", "strangulated"..., which appeared in several mentions of violence in CRIS dataset.

VERSION 2 – REVIEW

REVIEWER	Trevor Cohen University of Washington, Department of Biomedical Informatics and Medical Education
REVIEW RETURNED	07-Nov-2021

GENERAL COMMENTS

This is the first revision of a paper describing the application of transfer learning based NLP methods to identify mentions of interpersonal violence in electronic health record data. This is clearly an important application area for such methods, and the paper documents impressive NLP performance for the majority of violence-related constructs developed by the authors.

The revised manuscript is responsive to reviewer critiques in several respects. The authors have provided additional details on their use of word embeddings to expand a manually defined set of keywords (or keyword stems). This mitigates to a degree the concern that performance of the model on descriptions of the modeled interpersonal violence constructs that do not use any of the selected keywords remains unknown. The revision also includes key methodological details about construction of the test sets, justification for training multiple single-label models, and choice and weighting of the loss function for BERT.

Perhaps most importantly the authors define the scope of the paper as a validation of an NLP model that was selected on the basis of prior research, rather than a comparative evaluation of alternative approaches. This seems reasonable for a clinically-oriented journal such as BMJ Open, and as the chosen model may already have hit a ceiling in performance it seems unlikely that alternative models would perform better on the task as defined.

However, this does still leave some important questions unanswered. Even though the advantages of BERT-based models over SVMs (and LSTMs, when similarly configured) are apparent in the prior work the authors now discuss, one wonders whether or not additional fine-tuning on clinical notes (rather than the literature as with BioBERT) would be of benefit. After all, clinically fine-tuned models are publicly available also (e.g. BlueBERT, ClinicalBERT). Conversely, it may be the case that fine-tuning on the biomedical literature offers little benefit beyond the base BERT model, as the language in which interpersonal violence is described may already be well represented in BERT's original training corpus. In addition, while the full text CRIS work that is referenced was not accessible at the link provided, the title suggests it concerns occupation extraction, rather than violent incidents. So it may yet be the case that a simple baseline model performs well on this particular task, which would be of interest to readers without ready access to GPU resources (such resources should be acknowledged in the paper if they were used for fine-tuning in the current work, which seems likely given the training times provided in the response). While the authors' argument that comparative evaluation is beyond the scope of the current work seems reasonable, the inability of this work to address these methodological questions should be acknowledged as a limitation in the paper.

The newly-added statistics show excellent agreement between BioBERT and the annotators (Kappa of ~96%- ~98% across the six labels). In fact, this comfortably exceeds the inter-rater agreement described in the paper (Kappa of 60-85% for the six labels. It isn't clear by what metric "(i)nter-annotator agreement was high 82-96%"). Some additional explanation is required to resolve this discrepancy. It seems strange that model-rater agreement would exceed rater-rater agreement to this degree. Also, Table 5 is entitled "Kappa agreement between manually and automatically assigned categories in training and testing sets", but the agreement in the

	training set doesn't seem particularly relevant. Table 6, "Model error analysis" has two columns: "Model error analysis" and "False positives". However there is no explanation for what the "Model error analysis" column contains, and the false positives as a proportion of these seem higher than what one would anticipate given the performance statistics. While the additional details on the construction of the test set are appreciated, it is still difficult to understand where exactly this fits into the picture. It appears that 1411 sentences were extracted from the CRIS database using the 27 keywords, and that this set was manually annotated. However, in the results "a separate blind test consisting of 1000 newly annotated sentences" is described. In Table 2 we have "a blind test set with a 90% probability threshold (100 sentences). If these are different sets, they should be named differently in the text. The authors' response suggests that they may have been used at different stages (fine-tuning vs. development vs. blind testing). It would be helpful to explain in the text exactly how each of these sets were used. In summary, the revised manuscript is largely responsive to reviewer critiques. A few remaining concerns could be addressed by:  (1) Acknowledging unanswered methodological questions as limitations (2) Providing additional information (e.g. by expanding the captions) to aid interpretation of the newly-added Tables 5 and 6. (3) Explaining the discrepancy between system-rater and rater-rater agreement (the latter seems much higher, but one would expect the former to provide a ceiling on performance) (4) Clearly identifying the roles of the various "blind" test sets.
--	---

VERSION 2 – AUTHOR RESPONSE

Reviewer: 1

Dr. Trevor Cohen, University of Washington

Reference	Feedback	Action taken
1	Acknowledging unanswered methodological questions as limitations	This is an important limitation, and we appreciate it being raised. We have updated the limitations section to reflect that fine-tuning on clinical literature may provide better outcomes, and the lack of comparative evaluation within this paper limits the replicability of this work by others who may not have access to GPU resources.
2	Providing additional information (e.g. by expanding the captions) to aid interpretation of the newly-added Tables 5 and 6.	Thank you for this comment. Table 5 and 6 have been expanded upon within the text and should now be easier to interpret, and Table 6 has had an additional column added for clarity.
3	Explaining the discrepancy between system-rater and rater-rater agreement.	The likely cause of the discrepancy is that the model was trained on adjudicated data, whilst the disagreements between adjudicators (after rules were finalised) were largely due to inattention mistakes or unclear statements (which generally ended up classified as "irrelevant"). Additionally, the model-

		annotator agreement was computed on training and testing datasets, which would generate higher statistics than the blind test set.
4	Clearly identifying the roles of the various "blind" test sets.	We appreciate that clarification regarding tests would help readability of the paper. The roles of various sets have now been made clearer within the text throughout; a 3771 sentence training and test set used for 10-fold cross validation, a 1411 sentence additional test set, and a "blind" test set of 100 sentences by author AK to review sample generation, annotation and model assessment. We regret the initial error where the 100-sentence set was erroneously described as "1000" in a single fragment of text, and this has been corrected.

VERSION 3 – REVIEW

REVIEWER	Trevor Cohen University of Washington, Department of Biomedical Informatics and Medical Education
REVIEW RETURNED	15-Dec-2021

GENERAL COMMENTS	The authors have addressed all of my concerns in the revised manuscript, and are to be congratulated for the progress they have made in applying NLP to this important problem domain. I have no further critiques to add.
--